# Clinical Utility of Multiplex Ligation-Dependent Probe Amplification in the Genetic Assessment of Patients with Myelodysplastic Syndrome

**DOI:** 10.3390/biomedicines13122985

**Published:** 2025-12-05

**Authors:** Radostina Valeva, Maria Levkova, Dinnar Yahya, Mari Hachmeriyan, Ilina Micheva

**Affiliations:** 1Department of Medical Genetics, Faculty of Medicine, Medical University of Varna, 9002 Varna, Bulgaria; maria.levkova171@gmail.com (M.L.); dinnar.yahya@mu-varna.bg (D.Y.); mari.hachmeryan@mu-varna.bg (M.H.); 2Laboratory of Medical Genetics, UMHAT St. Marina, 9010 Varna, Bulgaria; 3Second Department of Internal Diseases, Faculty of Medicine, Medical University of Varna, 9002 Varna, Bulgaria; ilina.micheva@mu-varna.bg

**Keywords:** karyotype, myelodysplastic syndrome, MLPA, genetic variations, genetic analysis, prognosis, survival

## Abstract

**Background/Objectives**: Genetic abnormalities are critical for the diagnosis, prognosis, and therapeutic management of myelodysplastic syndromes (MDS). This study aims to evaluate the clinical utility of Multiplex Ligation-dependent Probe Amplification (MLPA) as a rapid and cost-effective method, determining its place alongside Next-Generation Sequencing (NGS) for the initial genetic assessment of patients with MDS. **Methods**: Bone marrow samples from 68 patients newly diagnosed with MDS were analyzed. Genomic DNA was investigated using the SALSA MLPA P414-C1 MDS probe mix to detect common copy number variations (CNVs). **Results**: MLPA detected genetic variants in 25 patients (36.8%). The most common finding was a single chromosomal abnormality (26.5%). Multiple pathological findings were observed in only 1.5% of patients, and a *JAK2* mutation was observed in 8.8% of the cohort. However, the presence of these aberrations did not show a statistically significant association with overall survival (OS) in the cohort. Patient sex was identified as the only variable that was associated with a marginal level of statistical significance regarding OS, indicating a worse prognosis for males. **Conclusions**: MLPA is a valuable, rapid, and cost-effective tool for initial genetic screening in low-resource settings. This was highlighted by our finding that sex was the sole significant prognostic factor, while the MLPA-detected variants were not found to be significant. The findings suggest that comprehensive risk stratification aligned with international standards requires more advanced molecular technologies.

## 1. Introduction

Myelodysplastic syndromes are a heterogeneous group of clonal hematopoietic stem cell neoplasms characterized by ineffective hematopoiesis, persistent peripheral blood cytopenias, and morphologic dysplasia [1]. A defining feature is the presence of fewer than 20% blasts in the bone marrow, which distinguishes MDS from acute myeloid leukemia (AML), although there is a variable risk of transformation into AML [2]. At the molecular level, these syndromes arise from the accumulation of somatic genetic and epigenetic alterations that disrupt normal hematopoiesis [3]. Therefore, the identification of these specific genetic events—particularly chromosomal abnormalities and gene variations—is not only essential for diagnosis but is also foundational for modern risk stratification and the prediction of therapeutic response [4,5].

Method selection is influenced by factors such as available resources at diagnostic facilities, patient volume, recruitment strategies, equipment, funding, and potential outsourcing. In countries like Bulgaria, the commonly preferred NGS and single-nucleotide polymorphism (SNP) array [6] are not supported by national healthcare policies and remain inaccessible for patients with hematological malignancies. Therefore, another reliable and cost-effective method is necessary, especially for smaller centers and populations, since processing fewer samples is more time- and resource-intensive.

MLPA is a semiquantitative polymerase chain reaction-based technique that simultaneously identifies various changes across many genomic regions. MLPA’s diagnostic scope encompasses CNVs, single-nucleotide variations (SNVs), and methylation changes. It is characterized by short and automated performance, accessible technical requirements, and cost-effectiveness. In addition, using pre-selected, synthetically synthesized primers and sex-matched healthy control samples in each reaction ensures MLPA’s high specificity [7,8,9]. This method is recognized as an alternative in the routine diagnostic process for various diseases, including hematological malignancies [10,11,12,13,14,15,16]. However, it is not commonly utilized in MDS cases, as indicated by the limited number of available published studies [7,8,17,18,19].

This study aims to assess the utility of MLPA as a tool for the initial genetic evaluation of patients with MDS, particularly in instances of limited coverage in terms of national health benefits.

## 2. Materials and Methods

Bone marrow samples were obtained from newly diagnosed patients from the Clinical hematology clinic of University Hospital “St. Marina”, Varna, for the period of one year (between July 2024 and July 2025) prior to the initiation of therapy. Genomic DNA was extracted using the Generi Biotech Extraction Kit (Generi Biotech, Prague, Czech Republic). A total of 12 nanograms of genomic DNA were utilized per MLPA reaction. We performed MLPA analysis with the SALSA MLPA P414-C1 MDS probe mix (MRC-Holland, Amsterdam, The Netherlands) [20]. This probe mix includes 46 probes that assess target specific genomic loci commonly involved in MDS, along with 12 internal control probes targeting genomic areas typically stable in MDS cases (Appendix A).

The MLPA procedure, including both negative controls and internal quality checks, was executed following the manufacturer’s recommended protocol. Amplified products were subsequently analyzed using the GenomeLab GeXP system (Sciex, Framingham, MA, USA). The interpretation of MLPA results was based on the calculation of the final ratio (FR), which is the normalized signal intensity of a target probe divided by the average signal of the internal control probes. The copy number status was determined using the following specific cut-off values, as defined in the SALSA MLPA P414-C1 user manual: normal copy number was assigned when 0.80 < FR < 1.20; heterozygous deletion when 0.40 < FR < 0.65; heterozygous duplication/gain when 1.30 < FR < 1.65; and homozygous deletion when FR = 0 [20]. Data analysis was carried out with the Coffalyser.net software version 240129.1959 (MRC-Holland, Amsterdam, The Netherlands), as per the supplier’s guidelines. 

We also collected and compared additional results from conventional cytogenetic analysis (CCA) for the selected patients in the study. In this case, a standard laboratory protocol for cancer cytogenetics was followed [19]. Bone marrow aspirates were collected at the time of diagnosis into sterile transport tubes containing the sodium heparin anticoagulant. Samples were processed within 24 h of collection. The prepared slides were aged (baked at 90 °C for 30 min) to stabilize the chromosome morphology. G-banding was achieved using a standard trypsin digestion followed by Giemsa staining (GTG-banding). Analysis was performed using a light microscope (Zeiss, Jena, Germany). A minimum of 20 metaphase spreads were fully analyzed for each patient, where available. A clonal abnormality was defined as the presence of at least two cells with the same structural abnormality or an extra chromosome, or at least three cells with the same missing chromosome. All results were described according to the International System for Human Cytogenomic Nomenclature **(ISCN 2020)** [21,22].

The analysis of genetic aberrations was limited to CCA and MLPA. Due to the initial design and available resources, NGS analysis was not performed on the samples in this cohort.

Overall survival was evaluated from the date of diagnosis until either patient death or the most recent follow-up. Survival data was analyzed using Kaplan–Meier survival curves, and differences between groups were determined by the log-rank test. Patient groups for Kaplan–Meier analysis were defined by two primary stratification variables: sex (male vs. female) and MLPA-defined genetic status (presence of any MLPA-detected abnormality vs. no MLPA-detected abnormality). A *p*-value below 0.05 was considered indicative of statistical significance. All statistical analyses were performed using the IBM SPSS Statistics software (Version 27).

## 3. Results

A total of 68 patients, diagnosed with MDS, were included in this study. The patient cohort consisted of 37 males (54.4%) and 31 females (45.6%), with a male-to-female ratio of approximately 1.19:1. The median age at the time of diagnosis was 73 years, encompassing an age range between 14 (one patient at 14 years old was also included) and 89 years.

As a further factor, genetic analysis using MLPA revealed that 43 patients (63.2%) displayed a normal genetic profile. Variants were detected in the remaining 25 patients (36.8%) accounting for 26 total genetic abnormalities (Table 1). Among those with abnormalities, the most frequent finding was an abnormality in a single chromosome, which was observed in 18 patients (26.5%). This includes predominantly clinically significant variants in chromosomes 5 (Figure 1) and 8. Two distinct MLPA-detected CNVs affecting different chromosomal loci were identified in a single patient (1.5%): a deletion on the long arm of chromosome 5 (del(5q)) and a complex deletion involving both arms of chromosome 7 (del(7p/q)) (Appendix A). Furthermore, a specific monogenic *JAK2* mutation (p.V617F) was detected in 6 patients (8.8%) (Figure 2).

However, when karyotyping was assessed, no metaphases were detected in 20 cases, indicating a high failure rate (29.4%). All of them were successfully analyzed by the DNA-based MLPA method (Appendix A). In another 4 cases, no cytogenetic analysis was performed (Appendix A).

Successful analysis was conducted in 44 patients, which comprised 64.7% of the cohort. The results of thirty-six of the patients that completed the analysis showed concordance with the results from CCA and MLPA analyses. This subgroup consisted of 26 normal results and 10 patients with confirmed pathological findings. The 10 pathological findings included five deletions in the 5q loci, three duplications in chromosome 8 (dup(8)), one deletion in 20q (del(20q)), and one duplication in the Y chromosome (Y-gain) (Appendix A). Only eight cases displayed non-concordant results between the two methods. We classified these differences into two main categories: technical difference and true methodological discordance in CNV detection. The technical difference category included two cases that were positive for *JAK2* somatic monogenic mutation (detected by MLPA). This reflects a difference in assay scope, as conventional cytogenetics is not designed to detect point mutations. True methodological discordance in CNV detection was observed in six cases, displaying disagreement in terms of detecting CNVs. This discordance comprised five cases where structural chromosomal abnormalities were detected by CCA but were missed by the targeted MLPA probes (specific karyotypes included missing Y chromosome (46,X,-Y), two cases of trisomy (47,XY), a hiphodiploid karyotype (40–44,XY), and one case of 45,X,-Y), and one case where the pathogenic CNV (dup(8)) was detected by MLPA, but the CCA result was normal, demonstrating the higher resolution of MLPA in that instance.

According to the IPSS-R (Revised International Prognostic Scoring System), the mentioned findings in chromosomes 5, 11, and 20 are associated with good prognosis when identified as isolated chromosomal aberrations. Conversely, the variants in chromosomes 8 and Y are reported as bearers of intermediate risk. Poor prognosis is only reported in the case of deletions in chromosome 7 and in one patient with two distinct variants [4]. The V617F status of *JAK2* is suggested to identify the RARS-T (refractory anemia with ringed sideroblasts associated with marked thrombocytosis) subtype of MDS [24], which was shown to correlate with better prognosis [21]. However, *JAK2* and other monogenic variants are not included in the IPSS-R classification. *JAK2* is also scored as favorable by the IPSS-M (Molecular International Prognostic Scoring System) [22].

To determine the prognostic significance of various factors, OS was evaluated, with death from any cause defined as the primary endpoint. We also considered the fact that all of the included patients have received treatment after the MLPA results were provided. The impact of sex and the presence of genetic aberration on patient survival were assessed separately. The analysis was performed using Kaplan–Meier estimates, the log-rank test, and Cox proportional hazards regression.

First, the influence of sex on survival rates was assessed by comparing males (*n* = 37) and females (*n* = 31) (Figure 3). The Kaplan–Meier survival curves for males and females showed a difference in survival probabilities, with the one-year survival rate being 89.2% for males and 100% for females. The median survival time could not be determined for either group because the survival curves did not drop below 50% during the observation period. This indicates that the median survival for both groups exceeds the maximum follow-up time (i.e., more than 50% of the subjects survived beyond the observation period). The log-rank test showed a *p*-value of 0.050 (chi-square = 3.832). This result represents a marginal level of statistical significance, suggesting a trend towards a difference in OS outcomes. This suggests that males had a higher risk of death compared to females, confirming that sex was a notable prognostic factor for survival in the selected study population.

In the next step, the association between the presence of genetic variation and OS was investigated (Figure 4). The patient cohort was divided into two groups: those without detected variants (*n* = 43) and those with at least one aberration (*n* = 25). The median survival time was undetermined for both groups, as the survival curves did not drop below 50% during the observation period. The log-rank test confirmed that there was no statistically significant difference in survival between the groups (*p* = 0.576). This indicates that the presence of genetic aberration, as detected in this study, was not a significant predictor of OS.

## 4. Discussion

The primary objective of this study is to evaluate the place of MLPA in the diagnostic and risk stratification algorithm in patients with MDS [25]. In this context, comparison with CCA is of paramount importance. Our findings demonstrate a significant practical benefit of MLPA: it represents a cost-effective, rapid and accessible alternative, especially in resource-limited settings. This benefit was clearly demonstrated in our cohort, where conventional karyotyping had a high failure rate (29.4%) due to failed cell culture or other causes (insufficient cellularity, coagulation, etc.). This is a well-known limitation of the method [26,27], and highlights the advantage of MLPA: as a DNA-based method, it does not require cell culture and successfully provided a result in all 24 patients in whom CCA failed or was not performed. This further supports its utility in high-risk scenarios.

When we evaluated MLPA as a stand-alone prognostic tool, our results regarding the patient genetic profile were not statistically significant. MLPA-detected variants did not indicate a significant association with OS. However, this apparent lack of prognostic significance is not necessarily a failure of the method, but rather highlights its defined limitations. While MLPA is highly efficient in detecting large CNVs, and specialized kits may include probes for specific point mutations (e.g., the *JAK2* p.V617F mutation in our assay), the P414-C1 MDS kit does not cover the full panel of key prognostic point mutations (e.g., *TP53*, *ASXL1*, *RUNX1*) [20,28], which are essential for modern risk scoring systems such as IPSS-M [29,30,31]. This is determined not only by the essence of the method but also by the specifications of the probe mix and the genetic markers included.

Furthermore, a notable finding in our study was that sex was the only variable that reached a marginal level of statistical significance concerning OS (*p* = 0.050). Worse prognosis was demonstrated in males (1-year survival 89.2% vs. 100% in females), which is consistent with data from large registries recognizing male sex as an independent negative prognostic factor [8]. Notably, all mortality events in the male subgroup in our cohort occurred in patients with a normal MLPA genotype. This strongly suggests that the risk in this key subgroup is determined by factors other than CNVs (which MLPA detects) and is likely related to point mutations, which some studies have reported to be more common in men [32].

Taken together, the results place the benefits of MLPA in a clear practical context. Our study confirms the role of MLPA as a reliable and essential diagnostic tool, especially when conventional karyotyping fails. However, our most important finding—that patient sex demonstrated a marginal level of statistical significance (*p* = 0.050) in predicting survival, a factor completely independent of the variants detected by MLPA—serves as strong confirmation of the move to multi-tiered molecular diagnostics.

The study presented has several restrictions. The main limitation is the reliance on MLPA as a single genetic method, which excludes the detection of prognostically important point mutations and makes it unsuitable for assessment of minimal residual disease (MRD) [33]. Second, the retrospective design and small sample size limit our ability to perform subgroup analyses and control for treatment-related confounding variables. Finally, the MLPA kit is labeled for research use only (RUO) [34], although its reliability has been demonstrated in other studies [9,35] and was supported by strict internal controls in our experiment.

The findings of our study, combined with the established role of sequencing-based diagnostics, emphasize the need to understand the practical strengths and weaknesses of each genetic method used in MDS (Appendix A). However, due to the study’s initial design and available resources, none of the samples in our cohort were analyzed by NGS, thus preventing a direct benchmarking of concordance and fidelity between MLPA and NGS. We recognize this limitation; in fact, the inability of MLPA to detect all key prognostic point mutations—which NGS covers—is a central conclusion of our study. Our findings highlight that MLPA’s primary value is in detecting CNVs and overcoming CCA failure, and it ultimately cannot replace NGS in modern, multi-tiered molecular diagnostics, needed to capture the full spectrum of risk point mutations, which appear to be particularly relevant in the male patient population.

## 5. Future Directions

Beyond the context of MDS, MLPA represents a valuable molecular diagnostic tool applicable to several other hematological malignancies. In chronic lymphocytic leukemia (CLL), specific MLPA kits are used to detect key prognostic CNVs such as deletions of 13q, 11q, and 17p, and trisomy 12, which are integral to disease staging and therapeutic decisions [36]. For multiple myeloma (MM), MLPA can reliably identify significant cytogenetic aberrations, including del (17p) and gain (1q), which are associated with high-risk disease [37]. Furthermore, in AML, especially those cases evolving from MDS, MLPA remains highly relevant for detecting shared chromosomal abnormalities (e.g., del(5q) and +8) [38]. This wide-ranging applicability reinforces MLPA’s position as a reliable and economical method for initial genetic assessment in a broad spectrum of hematological disorders.

Stemming from the findings of this study, several future directions for research and clinical implementation can be defined. The integration of NGS is a step towards a multilevel diagnostic approach to overcome the prognostic limitations of MLPA as a stand-alone method. A workflow can be established where MLPA serves as an initial, rapid screening tool for common CNVs in resource-limited settings. Patients with a negative or uninterpretable MLPA result should then be evaluated with more comprehensive, targeted NGS panels to identify the prognostically critical point mutations that influence survival [39,40,41,42].

The performance of future studies should be conducted prospectively and should integrate detailed patient treatment data. This will empower a more accurate assessment of the potential confounding effects of therapy on survival outcomes, providing a clearer picture of the true prognostic value of the genetic variants.

The outcomes of this study, the high significance of the patient sex on the one hand, and the lack of prognostic significance for genotype on the other hand, should be examined and validated in larger cohorts to ensure the statistical relevance and confirm whether these results are specific to the studied patient population or a broader phenomenon in similar background.

## 6. Conclusions

In the studied cohort of 68 patients with MDS, MLPA successfully identified genetic variants in 36.8% of patients, confirming its utility for detecting large-scale genomic alterations. Upon prognostic analysis, patient sex was found to have a marginal level of statistical significance regarding OS (*p* = 0.050). In contrast, the MLPA-detected variants (when analyzed collectively) did not demonstrate a statistically significant association with OS. These results are likely due to the limited coverage of MLPA with respect to prognostically critical point mutations, which are considered key factors in current risk stratification systems.

Clinically, MLPA remains a rapid, accessible, and valuable tool for primary screening of common chromosomal alterations in resource-limited settings. However, a negative or seemingly low-risk MLPA result should not be considered definitive for risk stratification, as its prognostic power as a stand-alone method is limited in modern molecular medicine.

## Figures and Tables

**Figure 1 biomedicines-13-02985-f001:**
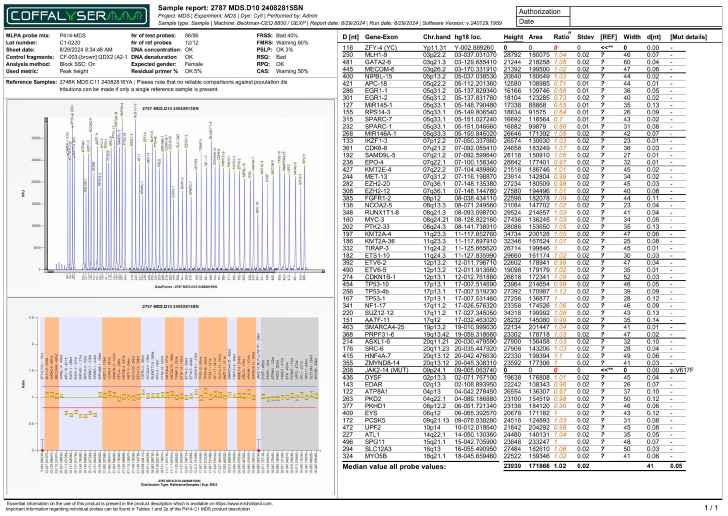
MLPA report (electropherogram - top panel, and ratio chart - bottom panel) displaying decreased ratios for targeted loci in chromosome 5, associated with deletions in those regions. Note: The horizontal red line indicates the 0.8 ratio threshold. The horizontal blue line indicates the 1.2 ratio threshold. The yellow dots indicate a pathological change in CNV, where the probe ratio falls below the expected normal range (0.8–1.2). The decrease in signal ratio for the 5q loci confirms a heterozygous deletion (loss of one copy). For *JAK2* p.V617F probe, the red dot in this specific MLPA report indicates that the mutation was not detected in the sample. Colored numbers represent the normalized signal intensities of the test probes and control probes. Technical flags (**) denote control probes (such as ZFY-4) that fall outside the standard reference range, often due to the patient’s sex (e.g., zero signal for the Y chromosome in a female patient). The numerical formatting and layout within the report image are the original, unmodified output from the Coffalyser.net software.

**Figure 2 biomedicines-13-02985-f002:**
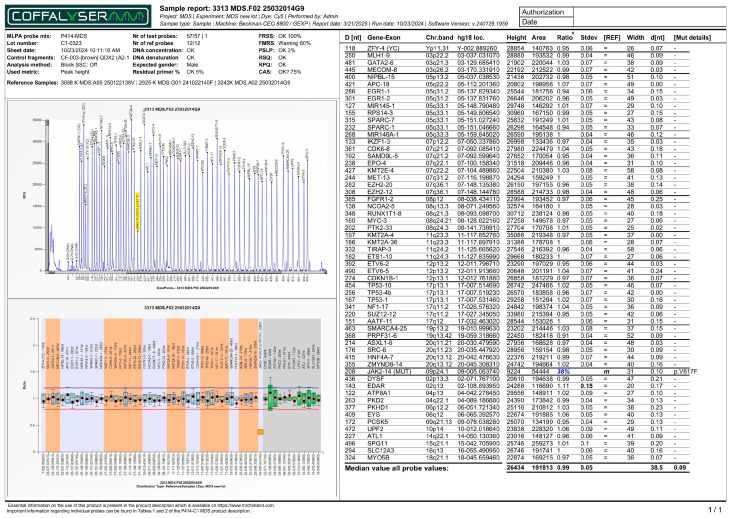
MLPA report (electropherogram - top panel, and ratio chart - bottom panel) of a patient with *JAK2* p.V617F monogenic variant (38%). All other genetic loci in the bottom panel display normalized ratios clustered around a normal value of 1.00. Note: The horizontal red line indicates the 0.8 ratio threshold. The horizontal blue line indicates the 1.2 ratio threshold. For *JAK2* p.V617F probe, the orange square in this specific MLPA report indicates that the mutation was detected in the sample.

**Figure 3 biomedicines-13-02985-f003:**
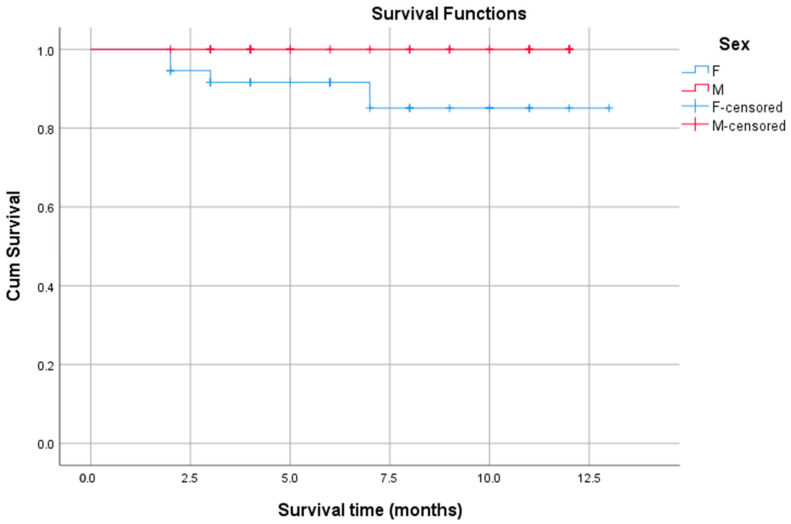
Kaplan–Meier survival plots stratified by patient sex. Note: The blue line (F) represents the survival function for the female sex group. The red line (M) represents the survival function for the male sex group. The blue “+” marks (F-censored) mean female patients were part of the study, but not a single female patient in this dataset experienced the event (e.g., death). The red “+” marks (M-censored) denote the male patients who were censored (i.e., did not die) at that particular point in time, distinguishing them from the drops in the red curve, which represent the male patients who did experience the event (death).

**Figure 4 biomedicines-13-02985-f004:**
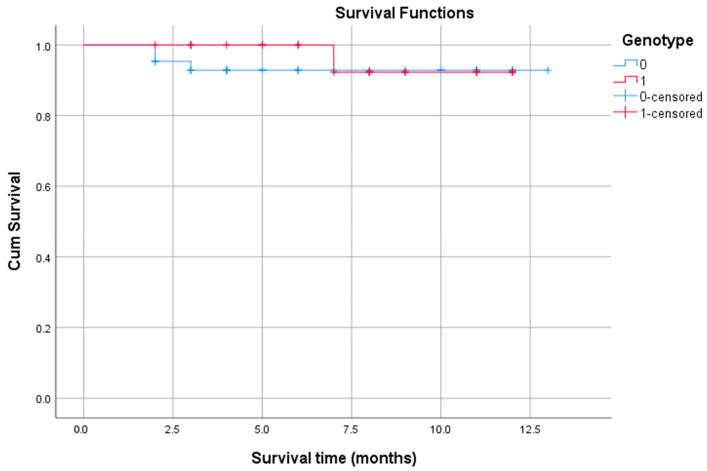
Survival plots based on patient condition. Note: The blue line represents the cumulative survival of patients in group 0 (with normal genotype). The red line represents the survival of patients in group 1 (with aberrant genotype). The blue “+” and the red “+” marks denote “censored” events for the patients in group 0 and group 1, respectively—one who was still alive at that specific time point. The drops in the curves represent the patients who did experience the event (death).

**Table 1 biomedicines-13-02985-t001:** Summary of MLPA-detected CNVs and *JAK2* p.V617F point mutations.

Genetic Variant	Number of Cases	Percentage of Patients	IPSS-R Cytogenetic Risk Classification *
del(5)	10	14.7%	Good
*JAK2* p.V617F	6	8.8%	-
dup(8)	5	7.4%	Intermediate
del(7)	2	2.9%	Poor
del(11)	1	1.5%	Good
del(20)	1	1.5%	Good
Y-gain	1	1.5%	Intermediate

Note: The total number of abnormalities listed (*n* = 26) exceeds the number of patients with an aberrant genotype (*n* = 25) because one patient had two distinct copy number variations (CNVs) that were counted separately. * IPSS-R Cytogenetic Risk Classification [23].

## Data Availability

Research data is unavailable due to privacy or ethical restrictions.

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
