# Peer review of "Clinical Utility of Multiplex Ligation-Dependent Probe Amplification in the Genetic Assessment of Patients with Myelodysplastic Syndrome"

_biomedicines, 2025, doi:10.3390/biomedicines13122985_

Round 1
Reviewer 1 Report
Comments and Suggestions for Authors
This study by Valeva et al describes the clinical utility of multiplex ligation-dependent probe amplification (MLPA) in assessing genetic aberrations of patients with Myelodysplastic Syndromes (MDS). The authors performed MLPA against genomic loci commonly involved in MDS and demonstrated that MLPA is robust in detecting a wide range of genetic abnormalities with good concordance with conventional cytogenetic assay.
While the findings are of interest to the broad audience, the paper would benefit from clearer articulation of the results and in-depth discussion, especially surrounding the technology of use. Comments and recommendations are provided below:
- Have any of the samples been profiled by next-generation sequencing? If so, can you conduct a benchmark between MLPA and NGS to test their level of concordance and fidelity?
- In reference to line 113, the author should provide a table (as supplementary material) that lists which 20 samples fail the conventional cytogenetic assays and whether they were successfully analysed by MLPA.
- The author should display karyotyping images of representative genetic lesions.
- In reference to line 127, can the authors discuss whether MLPA was able to identify patient with multiple variants?
- The author should discuss the advantages and limitations of MLPA, conventional cytogenetic assay, and the ‘gold standard’ next-generation sequencing in clinical diagnosis, and scenarios where MLAP may be a more favourable method. Possible factors of interest could be cost, turnaround time, accuracy, sample input requirement can be in the form of a table or part of Discussion.
- The author should discuss other haematological conditions that MLPA could be potentially applied.
- Line 105: The authors reported that variants were detected in 25 patients but Table 1 listed genetic abnormalities for 26 patients.
- Line 116: Did the author mean concordance?
- Line 123-125, list the relevant variants being discussed in these sentences.
- Legend for supplementary materials.
- Delete the extra space between ‘JAK’ and ‘2’ in Table 1.
- The legend of Figure 2 should be ‘MLPA report of a patient..”.
- The header of legend of Figure 4 should be ‘condition’. Replace ‘0’ with normal genotype and etc.
Author Response
Additional clarifications: The corrections are marked in red and underlined.
The numbers of the lines mentioned in some comments were different from the ones in my document for some reason. Apologies for the divergence, if there is any.

Reviewer 2 Report
Comments and Suggestions for Authors
The study entitled "Clinical utility of multiplex ligation-dependent probe amplification in the genetic assessment of patients with myelodisplastic syndrome" explores the value of MLPA in initial assessment of MDS patients and compares it against CCA. The work is conceptually relevant and methodologically acceptable (with certain caveats) and generally well-written, however, I noticed a number of issues that I believe should be addressed to enhance transparency and accuracy.
Major Comments
- In line 68, the amount of DNA used (12 ng) seems low for standard MLPA protocols. Most kits (including MRC-Holland's) typically require 20-100 ng per reaction. Please verify this value is correct.
- In line 70, listing two different LOT codes (C1-0220, C1-0323) for the probe mix is unclear. Specify if different lots were used for different samples/runs and, if so, state the rationale or timeframe for each lot's use.
- In line 78, the phrase "user manual criteria for cut-off values and final ratio of the probes" is vague. Briefly state the actual cut-off values used and clarify what "final ratio" refers to.
- In line 87, the aging method ("baked at 60°C overnight") is not standard practice in most modern cytogenetic labs. UV aging or chemical aging is relatively more common. Justify this specific method.
- In lines 94-97, specify how patient groups were defined for Kaplan-Meier analysis; were they stratified by MLPA-detected aberrations, IPSS-R risk groups or concordance/discordance between MLPA and CCA? Without this information, the analysis lacks context.
- IPSS-R assigns risk scores based on combined factors (cytogenetics, blast count, cytopenias), not individual variants. In Table 1, assigning "Good/Intermediate/Poor" to single abnormalities (e.g., del(11) = Good) could potentially be misleading. Please consider amending the table.
- In Table 1, JAK2 p.V617F is a point mutation, not a CNV detectable by standard MLPA. Its inclusion in an "MLPA-detected abnormalities" table may not be entirely accurate. Y-gain is often a constitutional finding, not somatic. Please review this table and verify whether the information is correctly mentioned.
- Counting JAK2 mutations (generally undetectable by CCA by design) as "differences" between methods is problematic to some extent. True discordance should only compare overlapping detectable abnormalities.
- MDS is rare in adolescents. The inclusion of a 14-year-old patient requires further justification, since this patient is technically considered an outlier.
- In lines 149-151, the statement "This means that 50% of the subjects in this group survived longer than that period" is incorrect. When median survival isn’t reached, it means >50% survived beyond the maximum observation time.
- Reporting 100% one-year survival in females versus 83% in males with a significant p-value (0.046) is statistically less probable with these sample sizes (31 females). Please re-evaluate the data and verify if this is correct to avoid overestimation and overinterpretation.
- Please kindly provide the timeframe of patient follow-up.
- In lines 192-195 (discussion), it is indicated that MLPA cannot detect point mutations, but Table 1 includes a point mutation as specified in earlier comments.
- The statistical significance of p=0.046 should be interpreted more cautiously in the discussion section, since it is a marginal level of significance.
Minor comments
- The reference list is mainly comprised of old publications. Consider citing more recently published relevant literature.
Author Response
The corrections are marked in blue and underlined.
The numbers of the lines mentioned in some comments were different from the ones in my document for some reason. Apologies for the divergence, if there is any.

Round 2
Reviewer 2 Report
Comments and Suggestions for Authors
I would like to thank the authors for addressing my comments. I have no further comments.